# Novel Design of Superhydrophobic and Anticorrosive PTFE and PAA + β − CD Composite Coating Deposited by Electrospinning, Spin Coating and Electrospraying Techniques

**DOI:** 10.3390/polym14204356

**Published:** 2022-10-16

**Authors:** Adrián Vicente, Pedro J. Rivero, Unai Urdiroz, Paloma García, Julio Mora, José F. Palacio, F. Javier Palomares, Rafael Rodríguez

**Affiliations:** 1Engineering Department, Campus de Arrosadía S/N, Public University of Navarre, 31006 Pamplona, Spain; 2Institute for Advanced Materials and Mathematics (INAMAT^2^), Campus de Arrosadía S/N, Public University of Navarre, 31006 Pamplona, Spain; 3INTA-Instituto Nacional de Técnica Aeroespacial, Área de Materiales Metálicos, Ctra. Ajalvir Km 4, 28850 Torrejón de Ardoz, Spain; 4Centre of Advanced Surface Engineering, AIN, 31191 Cordovilla, Spain; 5Instituto de Ciencia de Materiales de Madrid, CSIC, Cantoblanco, 28049 Madrid, Spain

**Keywords:** electrospinning, electrospraying, PTFE, PAA + β-CD, corrosion resistance, super hydrophobic, low water roll-off angle, adhesion resistance

## Abstract

A superhydrophobic composite coating consisting of polytetrafluoroethylene (PTFE) and poly(acrylic acid)+ β-cyclodextrin (PAA + β-CD) was prepared on an aluminum alloy AA 6061T6 substrate by a three-step process of electrospinnig, spin coating, and electrospraying. The electrospinning technique is used for the fabrication of a polymeric binder layer synthesized from PAA + β-CD. The superhydrophilic characteristic of the electrospun PAA + β-CD layer makes it suitable for the absorption of an aqueous suspension with PTFE particles in a spin-coating process, obtaining a hydrophobic behavior. Then, the electrospraying of a modified PTFE dispersion forms a layer of distributed PTFE particles, in which a strong bonding of the particles with each other and with the PTFE particles fixed in the PAA + β-CD fiber matrix results in a remarkable improvement of the particles adhesion to the substrate by different heat treatments. The experimental results corroborate the important role of obtaining hierarchical micro/nano multilevel structures for the optimization of superhydrophobic surfaces, leading to water contact angles above 170°, very low contact angle of hysteresis (CAH = 2°) and roll-off angle ($\alpha_{\mathrm{roll}-\mathrm{off}}\;$< 5°). In addition, a superior corrosion resistance is obtained, generating a barrier to retain the electrolyte infiltration. This study may provide useful insights for a wide range of applications.

## 1. Introduction

In recent years, nanocomposites have become very attractive materials with which to design different functional properties for a broad range of applications. The addition of nanoparticles to the polymeric material may lead to changes in their properties that could be interesting for their applications. Among them, superhydrophobicity (SH) has been attracting attention due to properties that make them suitable to use in numerous fields [1,2]. One of the most representative examples of these materials are polymeric coatings, to which the addition of PTFE nanoparticles significantly improves their hydrophobicity, low sliding (roll-off) angles, low hysteresis, immersion stability, corrosion resistance, low ice adhesion, high thermal conductivity, dielectric and mechanical properties, and corrosion resistance [3,4,5,6,7,8,9,10]. As a result, there is a wide range of interesting applications, such as self-cleaning, liquid transport efficiency, naval coatings, seaplanes, off-shore wind power and oil platform applications, among others [11,12,13,14,15,16].

PTFE is a thermoplastic semicrystalline fluoropolymer with excellent chemical resistance, high thermal stability, low-friction coefficient, and superhydrophobicity characteristics as a consequence of the strong bonds between the fluorine and carbon atoms. However, the high crystalline melting point (337 °C) and its insolubility in common solvents make PTFE difficult to process [17,18,19] and adhere to different surfaces [20,21]. These difficulties, coupled with the need to manufacture materials that combine high surface roughness with the PTFE low surface energy [22,23], have led to a number of different production techniques. Among them, we can include roughening with sandpaper, microetching processes, laser-plasma X ray, laser/femtosecond laser, 3D printing, laser ablation, hot embossing process, and spray coating among others [24,25,26,27,28,29,30,31]. Recently, these procedures have been joined by the electrodynamic techniques, electrospinning [32,33,34] and electrospraying [20,35], which are facile, cost effective, and flexible methods that utilize an electrically charged jet of polymer solution for different scales of fiber production [36]. In addition, another method compatible with PTFE coatings is spin coating. This technique uses the centrifugal forces created by a spinning substrate to spread a coating solution on a surface and obtain films with a given and homogeneous thickness [37,38].

In this work, a superhydrophobic PTFE coating with excellent water mobility and superior corrosion protection is proposed. In the preparation of this kind of coating, the adhesion of the PTFE particles to the coated surface becomes a key point that still needs to be optimized [20,21]. In order to maximize the hydrophobic properties and improve the coating adhesion, a three-step process combining electrospinning, electrospraying, and spin-coating techniques has been used. In the first one, a PAA + β-CD electrosprayed fiber membrane is deposited onto the substrates to take advantage of its adhesive properties [39,40,41]. In the second one, a PTFE spin-coated layer is deposited, thus obtaining a uniform particle distribution embedded in the porous PAA + β-CD network that leads to WCA values similar to those reported in literature for PTFE fibers [42,43,44,45]. Finally, an extra layer of electrosprayed PTFE particles leads to the formation of a high roughness, increasing the hydrophobicity with WCA values at the level of the best ones found in the literature [46,47]. In each step, samples were subjected to different heating treatments to improve PTFE adhesion to the coated surface leading also to a high corrosion resistance, making this material a good candidate for multiple outside applications. 

## 2. Experimental Procedure

### 2.1. Materials and Reagents

The AA6061-T6 aluminum alloy flat substrates were purchased from LANEMA de Aluminios y Plásticos (Madrid, Spain) with final dimensions of 50 mm long, 50 mm wide, and 2 mm thick. Substrates were polished and cleaned with acetone, obtaining a roughness lower than Ra = 0.8 µm. 

Poly(acrylic acid) (PAA, C_3_H_4_O_2_, Mw ≈ 450,000 g/mol), β-cyclodextrin (β-CD, C_42_H_70_O_35_, purity 98%) and absolute ethanol for analysis Emparta^®^ ACS (CH_3_CH_2_OH) were purchased from Sigma-Aldrich (Saint Louis, MO, USA). The aqueous PTFE dispersion (Teflon^®^ PTFE DISP 30, PTFE-1) was acquired from Chemours Fluoropolymers (Valencia, Spain). Dispersion contained 60 wt% of the 220 nm average size particles, 34 wt% of the distilled water, and 6 wt% of a non-ionic surfactant (polyoxyethylene alkylether).

### 2.2. Fabrication Techniques

#### 2.2.1. Solution and Dispersion Preparation

The PAA was dissolved in a mixture of ethanol and distilled water with a volume ratio 2:3 ($V_{\mathrm{Eth}}:V_{H_{2}O}$), resulting in a PAA solution of 5 wt%. Then, the β-CD was added in a weight ratio 3:5 ($w_{\beta -\mathrm{CD}\;}:w_{\mathrm{PAA}}$) of PAA. Thus, a homogeneous PAA and β-CD blend solution was obtained at room temperature and under stirring (200 rpm) for 72 h.

The PTFE dispersion (PTFE-1) was dissolved in an ethanol and distilled water mixture with a volume ratio 1:2 ($V_{\mathrm{Eth}}:V_{H_{2}O}$). Thus, a PTFE dispersion of 30 wt% PTFE nanoparticles (PTFE-2) was obtained at room temperature and under weak stirring (50 rpm) for 10 min.

#### 2.2.2. Electrospinning (ES), Electrospraying (SP) and Spin-Coating (SC) Processes

The main system used for this work was Yflow^®^ Professional Electrospinning Machine (Yflow^®^ S.D., Malaga, Spain). It is used for electrospinning and electrospraying, enabling a control on different operation parameters, such as Taylor cone visualization, flow rates, evaporation distance, electrical currents, and droplet diameter deposition. A double polarization applied voltage is used (needle/collector) to eliminate the nanoparticles “fly out” and improve both procedures. The positive electrode was a needle with an outer/inner diameter of 0.9/0.6 mm, and the negative electrode was a flat collector. Both processes were completed at room temperature 24 °C ± 3 °C and 28% ± 5% relative humidity (RH). In addition, a homemade spin-coating (up to 3800 rpm) system was developed to obtain homogeneous coatings with the PTFE-1 dispersion.

#### 2.2.3. Coating Fabrication Process

A three-step-process of electrospinning, spin-coating and electrospraying was used to prepare the superhydrophobic coatings onto the aluminum substrates. Table 1 and Figure 1 summarize the fabrication process stages with both the corresponding deposition techniques as well as heat treatments (HT_x_) at each stage.

*(i)* 
*Electrospinning + HT_1_*


In the first step (Figure 1i), the PAA β-CD blend solution was loaded into a 10 mL syringe located vertically on the electrohydrodynamic system and used for the preparation of an electrospun coating (S1) with the parameter configuration shown on Table 2 (ES).

After the electrospinning process, sample S1 was placed in a high-temperature muffle at 220 °C (HT_1_) for 40 min (S2) to remove the residual solvent, achieving a high degree of water insolubility and an improvement of the mechanical strength [39,40]. The change is produced by an increase of the crosslinking agent (β-CD) activation that functionalizes the PAA. This leads to the formation of the ester bonds in an esterification process.

*(ii)* 
*Spin Coating + HT_2_*


Once the S2 is cooled to room temperature, the coating was placed in the spin-coating machine (Figure 1ii). In a first step, 3 mL of the PTFE-1 aqueous dispersion was infused over the surface wetting the whole area. Then, the sample was accelerated by setting a speed of 3500 rpm for 2 min to remove the dispersion excess getting a quick drying. After that, the sample was placed in a high-temperature muffle at 260 °C (HT_2_) for 1 h (Sample S3) to remove the wetting agents. 

*(iii)* 
*Spin Coating + Electrospraying + HT_3_*


In the final step, the spin-coating procedure (PTFE-1) was repeated to obtain a homogeneous wet layer in which the electrosprayed PTFE particles (PTFE-2) may remain anchored by obtaining a hard binding. The parameters used during the electrospraying process are summarized on Table 2. Finally, a heat treatment at 340 °C (HT_3_) for 80 min is carried out (Sample S4).

### 2.3. Characterization Techniques

The thickness of the samples was obtained by measuring the step high of a vertical profile image, using a confocal microscope (model S-mart, Sensofar Metrology, Barcelona, Spain) and a contact profiler (Mitutoyo Surftest SV2000 N2, ISO 4287, resolution: 0.1 nm, cut off λc: 0.25 mm). 

On the one hand, the resultant surface morphology of the samples was analyzed by a field-emission scanning electron microscope (FE-SEM, Hitachi S4800, Tokyo, Japan). On the other hand, surface roughness was measured with an ALPA SM-Rt-70 contact profiler (ISO 4287, resolution: 0.001µm, cut off λc: 0.8 mm), and every sample was measured at five different parallel locations equally distributed approximately considering the average value. Moreover, sample roughness was also checked by means of Wyco RST-500 interferometric profilometer (Veeco Corporate, Plainview, NY, USA) by using the vertical scanning interferometry (VSI) mode. Three-dimensional (3D) images of the surfaces were obtained, and the corresponding changes in surface roughness average were measured.

In order to obtain chemical information about the samples, the present functional groups were determined by a Fourier-transform infrared (FTIR) spectroscopy study, by using a Perkin Elmer Frontier spectrophotometer (Waltham, MA, USA) in the spectra range of 600–4000 cm^−1^. To complete this information, an analysis of the elemental and chemical composition of PAA, β-CD, PTFE (reference samples), S1, S2, S3, and S4 set of samples was performed by X-ray photoelectron spectroscopy (XPS). These experiments were carried out in a UHV chamber with a base pressure of 10^−10^ mbar equipped with a hemispherical electron energy analyzer (SPECS Phoibos 150 spectrometer) and a 2D delay-line detector, by using an X-ray source of Mg-Kα (1253.6 eV) [48]. XPS spectra were acquired at normal emission take-off angle by using an energy step of 0.50 and 0.10 eV and a pass energy of 40 and 20 eV for survey spectra and detailed core-level regions, respectively. The surface charging effect built up upon the photoemission experiments, which produces both energy shift and line shape modifications of the spectra, has been compensated by using a low-energy electron flood gun. In addition, it is well known that extensive electron irradiation might produce some degree of chemical damage in polymeric samples in the sense that bonds are “broken”. An indication of the strength of bonding is then related to the stability or modification of the sample surface upon the exposure to the electron beam. The spectra were analyzed with the CasaXPS program (Casa Software Ltd., Cheshire, UK) by using a Shirley method for background subtraction and data processing for quantitative XPS analysis. Spectra are displayed after the subtraction of the contribution of the Mg-Kα satellite emission. The absolute binding energies of the photoelectron spectra were determined by referencing the sp2 transition of C 1s at 284.6 eV determined from a freshly cleaved highly oriented pyrolytic graphite (HOPG) sample.

A Scratch Tester (Scratch Tester Revetest, CSM instruments, Anton Paar GmbH, Graz, Austria) determined the coating adhesion to the substrate. For this purpose, linear scratching with a Rockwell type diamond indenter of 200 µm tip radius) and a progressive load from 1 N to 20 N (loading rate 100 N/min, speed 10 mm/min) was used.

Water contact angle (WCA) measurements were carried out with a CAM 100 contact angle goniometer (CAM 100, KSV Instruments, Burlington, VT, USA) by using 10 mL of distilled water at room temperature. The static water contact angle was measured three times and in six different places considering their average as the final value. In addition, the contact angle of hysteresis (CAH) and the sliding/roll-off angle ($\alpha_{\mathrm{slide}}/\alpha_{\mathrm{roll}-\mathrm{off}}$) was measured with an optical tensiometer (Attension Theta Lite, Biolin scientific, Gothenburg, Sweden) placing 10 mL of distilled water (DIW) at room temperature on the levelled surface to be measured, mounted on the C218 tilting module stage. The platform was manually tilted from 0° to 90° at an approximate rate of 3°/s. The measured values were taken at the inclination where the droplet started sliding or rolling.

Electrochemical measurements of Tafel polarization curves were carried out on an Autolab Potentiostat/Galvanostat PGSTAT302N (Metrohm, Herisau, Switzerland). All corrosion tests were performed at room temperature in 6 wt% NaCl aqueous solutions by using a conventional three-electrode cell consisting of a working electrode (bare or coated Al sample), a silver chloride Ag–AgCl reference electrode, and a platinum counter electrode. Before conducting all the experiments, the samples were immersed in the 6 wt% NaCl electrolyte for 30 min to make sure that the system was in steady state and with the open circuit potential (OCP) stabilized. The set-up employed for this measurement can be seen in Figure 1 of reference [49].

Tafel polarization measurements were obtained by scanning the electrode potential automatically from −150 mV to +150 mV with respect to the OCP voltage at a scan rate of 2 mV s^−1^.The output from these experiments yielded a polarization curve of the current density versus the applied potential. The resulting corrosion current can be calculated by using Tafel slope analysis, where it established a relationship between the current density and the electrode potential during the polarization. The corrosion data were obtained from Tafel polarization curves through the slopes of the anodic/cathodic parts (βa/βc) by superimposing a straight line on the linear portions of the cathodic and anodic curves. Finally, other corrosion parameters, such as equivalent weight of the metal, density, or exposed surface, are also required as input parameters. With this information, the software generates the complete set of corrosion parameters. Thus, the corrosion rate, corrosion current density (Jcorr), corrosion potential (Ecorr), and protection efficiency (η) were calculated according to the equations of reference [50].

## 3. Results and Discussion

### 3.1. Sample Thickness and Surface Morphology

Sample thickness was measured by using confocal image microscopy (S1, S2, and S3), where a step function of the cross-section vertical profile at a different location [51] was employed obtaining the average values. On the S4, a contact profiler was used as it was not possible to obtain an image with the confocal microscope. A step profile was obtained with the contact profiler at several locations, resulting in the mean values. In addition, samples’ fiber diameter and their structure were analyzed with FE-SEM. The surface roughness was studied by interferometric profilometry (S1, S2, and S3) and contact profilometry (S4) to understand the effect of the different procedures during their preparation. On S4, the contact profiler was applied, because it was not possible to obtain an image with the interferometric profilometer. The obtained values for the thickness, fiber diameter and surface roughness are summarized in Table 3.

The fiber diameters in the Figure 2a histogram are presented with diameter size for samples S1 (a) and S2 (b) having a log normal distribution shape, typically observed in nanoparticles’ size distribution that leads to the average values presented in Table 3. The observed fiber diameter decreases from S1 to S2 as is clear in the FE-SEM images shown on Figure 3a,b. As can be seen in Figure 3a there is a network of fibers with irregular sizes and shapes, where partial crosslinking occurs between the β-CD and PAA for the S1 sample. The first thermal treatment HT_1_ leads to the excedent solvent evaporation (ethanol and water) and an increase of the crosslinking agent (β-CD) activation with PAA. The rest of unreacted β-CD is detached from the fibers, decreasing their diameter; it then agglomerates and crystallizes in cubic shapes as can be seen in Figure 3b. The β-CD crystallization process has been extensively studied [52,53,54,55]; theform depends on the chemical compounds surrounding the polymer and the solvent used. The presence of the β-CD cubic crystals within the fiber network increases both the roughness and the sample thickness.

As for the rest of the samples, roughness has a huge decrease from S2 to S3, resulting in a smooth surface due to the SC_1_ + HT_2_ procedure. The FE-SEM images of Figure 3b,c show how the PTFE-1 covers uniformly the fibers of S2 and, as a result, a fairly uniform coating of PTFE nanoparticles is achieved for S3. On the contrary, the SC_2_ procedure of PTFE-1 with the SP of PTFE-2 dispersion increases surface roughness from S3 to S4. The PTFE particles of S4 (see Figure 3d) are randomly deposited and form strong bond with each other and with the PTFE particles of lower layers obtained from S3 + SC_2_ process during HT_3._ This results in a random network of spheres with a micro/nano multilevel structure through the combination of different size particles leading to a rough surface structure [20].

### 3.2. Chemical Composition 

In order to identify the chemical composition of the samples surface, FTIR and XPS studies were performed. In Figure 4, FTIR spectra of the electrospun samples S1 and S2 show no significant differences between them. They have absorption bands characteristics of ester bonds, produced by the crosslinking agent β-CD in the PAA solution by the esterification process. More specifically, a strong absorption band in the 1750–1700 cm^−1^ region is attributed to the ester carbonyls and to the carboxylic carbonyl bonds, present in both the free carboxylic acid groups of PAA and in the β-CD-PAA ester bonds. The bands in the 1300–1100 cm^−1^ region are attributed to the C-O stretching vibrations of carboxylic and ester functional groups and, the bands in the 1050–900 cm^−1^ region could be associated to the OH deformation mode of the alcohol groups of β-CD, which have not crosslinked with PAA fibers, giving evidence of a partial crosslinking reaction between the β-CD and PAA. Moreover, the O-H absorption peak shows a decrease for S2 as a result of an increase of the crosslinking agent (β-CD) activation with PAA after HT_1_ [39]. 

In the case of S3, the FTIR spectrum shows strong absorption peaks at 1210 cm^−1^ and at 1151 cm^−1^, which correspond to the C-F bond and C-C bond of the PTFE-1, respectively. The peak at 640 cm^−1^ could be attributed to the absorption of CF_2_ wagging [56]. As can be observed S3 presents the same contributions as the PTFE reference sample as expected because of the SC_1_ process. Finally, in the S4 sample, the FTIR spectrum is completely flat, reflecting light in the infrared region and avoiding the obtention of any information. To solve this and obtain complementary information on the samples surface, XPS studies were carried out. The overall sample composition was determined from survey spectrum which showed intense signals from C, F, and O. C, F, and O atomic concentrations were determined by measuring the integral areas of C 1s, F 1s, and O 1s spectra, taken at a pass energy of 20 eV, after background subtraction and normalization by using the sensitivity factors proportional to the Scofield cross-section provided by the electron energy analyzer manufacturer. S1 and S2 data are in agreement with those obtained by FTIR showing PAA and β-CD presence. However, some extra information was extracted from S3 and S4 in comparison with a reference PTFE sample. The elemental content of C and F of S3 and S4 correspond to the nominal atomic of PTFE (70% F; 30% C). From the quantitative analysis of XPS peaks upon sample processing under electron beam irradiation, it is observed that the F vs. C atomic content ratio is reduced from (initial 70–30% F, final 50–50%for PTFE). 

This fact is in line with the damage of CF_2_ bonds and the decrease of both the associated CF_2_ peak in the C 1s (Figure 5a) and the continuous loss of F 1s core level emission. In addition, this is also related to the increase signal of C 1s in the energy region of the photoelectrons from C-C, and the emission of C-OH and C=O groups, whose intensity also follows the trend of the O1s signal. Regarding our samples, XPS spectra of S3 and S4 exhibit the characteristic photoelectron emission of PTFE comprised of the C 1s and F 1s intense peaks centered at the binding energy of ca. 292 to 690 eV, respectively. Detailed peak shape analysis of C 1s emission were performed by the deconvolution of the C 1s spectrum with several Gaussian/Lorentzian symmetric components (ratio of 70/30) by using a least-squares fitting routine [48]. The energy position of the peaks and their relative heights were determined to account for the emission ascribed to the different chemical environment of carbon atoms) according to the values reported in previous works [57,58]. The result of the fit provides symmetric peaks from CF_2_, CF, CF_x_ representative of the PTFE nature of the samples and several components C=O, C-OH, C-C attributed to the presence of various oxygen functional groups that were present on the irradiated PTFE sample. Upon comparison with the line shapes, no significant differences are observed other than the intensity of the peaks for the oxygen functional groups as can be seen in Figure 5b. However, the O 1s emission signal is quite different for the PTFE, S3 and S4 samples as can be seen in Figure 5c. The PTFE sample shows a clear presence of O which, as we can see in Figure 5c, decreases for S3 and is nil for S4. This presence explains the differences discussed above and may be indicative of the hydrophobic qualities of the samples being S4 the better one. Regarding the bonding stability, PTFE reference sample shows a decrease for the CF_2_ signal giving way to the appearance of the secondary components mentioned as the irradiation time goes on. This phenomenon is smaller for S3 sample and non-existent for S4 suggesting that chemical bonds reduce their quality but are more stable for higher-temperature treatments.

### 3.3. Coating Adhesion Study by Scratch Test

The adhesion of the samples was evaluated by the scratch test to compare the adhesive failure and strength of the different coatings on the substrate. The S1 sample had poor adhesion, not good enough to be tested in the scratch test machine. The S2 sample did not show any scratch resistance, with gross delamination and adhesion failure from the early stages of the test (see Table 4). The S3 sample exhibits an improvement of scratch resistance compared to S2 due to the SC_1_ of PTFE-1 dispersion and HT_2_. This higher value is associated with the weak polyelectrolyte nature of the polymeric precursor of PAA, which can display a high density of carboxylic groups and negative charges, making possible a complex balance of interactions (mostly electrostatic and Van der Waals interactions) that can also promote a better adhesion onto the substrate [59]. In this way, the PTFE nanoparticles are fixed in the fiber matrix, resulting in a composite, wherein a higher cross-linking leads to higher scratch values [60]. Finally, the scratch resistance is increased in S4 compared to S3. This increase is mainly obtained by a strong bonding of the electrosprayed PTFE nanoparticles with each other and with the PTFE particles of lower layers during HT_3_, where the particles are melted and progressively cooled. The HT_3_ not only leads to particles bonding, but also improves the adhesion to the substrate.

In addition, the highest adhesion strength obtained in S4 (see Table 4) is comparable to scratch critical loads of Al/epoxy coatings (from 6 N to 12 N) [61]. According to previous work on electrosprayed PTFE coatings, one of the main problems is the adhesion of PTFE nanoparticles to the substrate [20,21]. The optical images of the representative scratches are presented in Figure 6. As can be seen, the surface trace of the samples exhibits a cleaner trace in those with lower critical loads, unlike those with higher critical loads, which show more tearing. Therefore, this work implies a remarkable improvement in the bonding of PTFE particles to each other and their adhesion to the substrate.

### 3.4. Wetting Properties

The PAA + β-CD coatings (S1 and S2) have shown a superhydrophilic behavior having different surface roughness (see Table 3). The S1 sample was not considered in this study because the water insolubility of the fiber mat was achieved by a thermal treatment (HT_1_), inducing a thermal crosslinking reaction between the PAA and the β-CD molecules. Thus, the fibers of S2 remain almost unaltered, showing high chemical and mechanical stability absorbing the water droplets [39,62]. In both cases, S1 and S2 show a superhydrophilic behavior with static WCA < 5°. 

However, in sample S3, the coating shows a complete change of behavior, exhibiting a clear hydrophobicity with a WCA = 142° ± 7° in spite of being the smoothest of all the samples (see Figure 7). The water repulsion of the fluoride functional groups previously mentioned fully explains this behavior [56,63].

On the other hand, the geometrical structure of the surface plays an important role in the static wettability properties. This effect can be clearly seen between samples S3 and S4, where the hydrophobicity of S3 is mainly caused by the chemistry and the SH of S4 (static WCA = 170° ± 4.2°) by the combined effect (chemistry and the geometrical structure), especially as a result of the remarkable increase in roughness due to the SP process of PTFE nanoparticles (see Figure 7).

In addition, the roll-off water angle ($\alpha_{\mathrm{roll}-\mathrm{off}}$) and the contact angle of hysteresis (CAH) of the S3 and S4 samples (see Table 5) were measured to check the dynamic wetting properties. The S4 sample obtained a higher performance through lower CAH and $\alpha_{\mathrm{roll}-\mathrm{off}}$ values. This implies a low water interaction generated by the electrosprayed PTFE nanoparticles. 

The significant improvement of the dynamic wettability properties obtained in S4 is mainly caused by the combination of the PTFE chemical repulsion and the multilevel roughness of the surface. This is induced by the formation of micro/nano-sized particles, resulting in a Cassie–Baxter state [64,65]. The presence of air entrapped inside the gaps particles, makes the penetration of the liquid into the roughness or texture of the surface, difficult due to capillary forces, leading to a superhydrophobic surface [66,67]. 

### 3.5. Anticorrosion Performance

In order to check the anticorrosion behavior, Tafel polarization tests were performed obtaining the polarization curves presented in Figure 8. First, the aluminum substrate (AA6061T6) was tested as a reference (Figure 8) to be later compared with the samples (S3 and S4). In this case, the sample S1 and S2 were not studied because of their superhydrophilic behavior, where there are almost no differences to the substrate. In polarization curves, the excellent corrosion resistance has a lower corrosion rate (CR), which is related to a lower corrosion current density (Jcorr) and a higher corrosion potential (Ecorr) [68,69,70]. The results show that all the samples minimize the corrosion rate and current density of the aluminum substrate (see Table 6). 

Furthermore, Table 6 indicates that an increase of superhydrophobicity and a low water adhesion strength are related to a considerable decrease in Jcorr and CR, leading to an improvement in the substrate protection efficiency (η).

This phenomenon could be due to a better blocking of the corrosion current because of water repulsion, which creates a barrier to retain the electrolyte infiltration to a larger extent. This effect is more intense in S4, where the tested surface was completely dry after removal of the electrolyte. However, in S3, the surface was wet as a consequence of the electrolyte infiltration. Thus, S4 and S3 minimize the corrosion rate of the aluminum substrate by four orders and one order of magnitude, respectively. Finally, these results are in concordance with reported literature [21,71,72], where PTFE coatings show a superior corrosion resistance, showing a great potential to be used in harsh acidic industrial environments. This effect is associated with the presence of a robust air layer entrapped on the superhydrophobic surface, which considerably reduced the contact area between coating and liquids, acting as an efficient barrier to prevent the penetration of aggressive media to the underlying metal substrate. 

## 4. Conclusions

In the present study, we have developed a composite PAA + β-CD + PTFE coating through the combination of ES, SC and SP techniques and successive heat treatments. The addition of PTFE nanoparticles in the PAA + β-CD electrospun mat turn the surfaces from hydrophilic to hydrophobic (WCA = 142°) because of the fluorine groups (CF_2_).

In order to produce superhydrophobic surfaces and with excellent water mobility, the SP of modified PTFE dispersion leads to water contact angles above 170° and very low contact angle hysteresis and roll-off angle (CAH = 2°, <5°). In this way, the FE-SEM images corroborate the important role of obtaining hierarchical nano/microparticle composite and the creation of micro/nano multilevel structures for the optimization of superhydrophobic surfaces. In addition, the superhydrophobic surface exhibits a significant improvement of the mechanical properties with a scratch resistance in the range of Al/epoxy coatings. This increase is mainly obtained by a strong bonding of the electrosprayed PTFE nanoparticles with each other and with the PTFE particles of lower layers fixed in the PAA + β-CD fiber matrix, where there are a high density of carboxylic groups and negative charges. Therefore, the different heat treatments (HTx) play a fundamental role not only for wetting agent removal, but for the improvement of particles bonding with each other and their adhesion to the substrate. Finally, the corrosion resistance shows an excellent performance, where a remarkable protection is achieved, blocking the corrosion current by water repulsion, which creates a barrier to retain the electrolyte infiltration. Thus, the surface remains dry with the electrolyte in the SHS surface and chemical inertness by the PTFE.

## Figures and Tables

**Figure 1 polymers-14-04356-f001:**
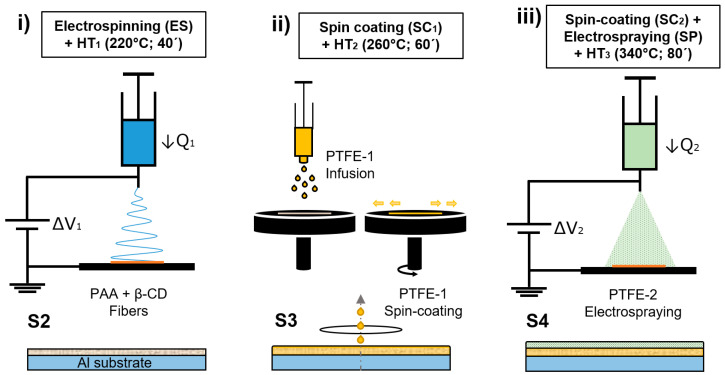
Schematic illustration of the fabrication methods to produce S2, S3, and S4 composite samples. (**i**) Electrospinning a PAA + β-CD fibrous coating and HT_1_. (**ii**) S2 + spin-coating with PTFE-1 dispersion and HT_2_. (**iii**) S3 + spin-coating with PTFE-1 dispersion, electrospraying with PTFE-2 dispersion and HT_3_.

**Figure 2 polymers-14-04356-f002:**
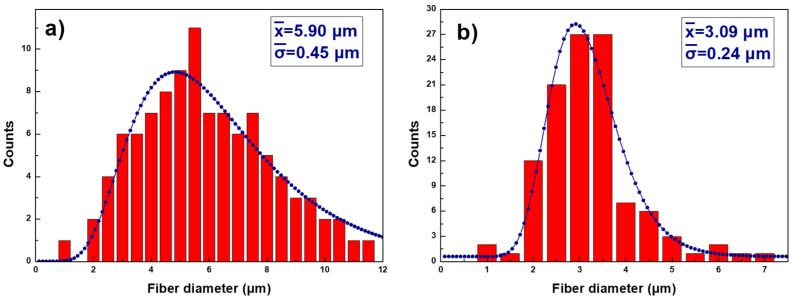
Histograms of the diameter distribution of the fiber samples S1 (**a**) and S2 (**b**).

**Figure 3 polymers-14-04356-f003:**
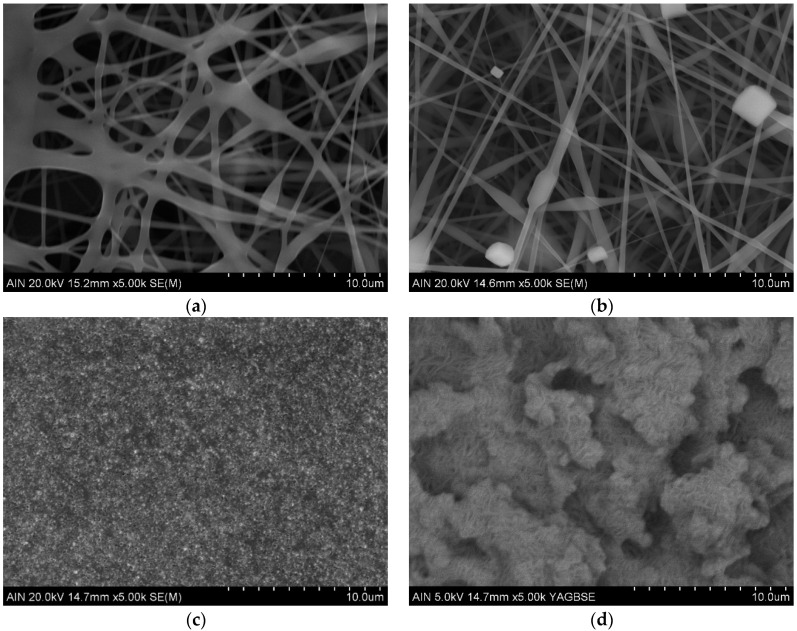
Field- emission scanning electron microscopy (FE-SEM) images of the samples surface morphology. S1 (**a**), S2 (**b**), S3, (**c**) and S4 (**d**) at the scale of 10 µm.

**Figure 4 polymers-14-04356-f004:**
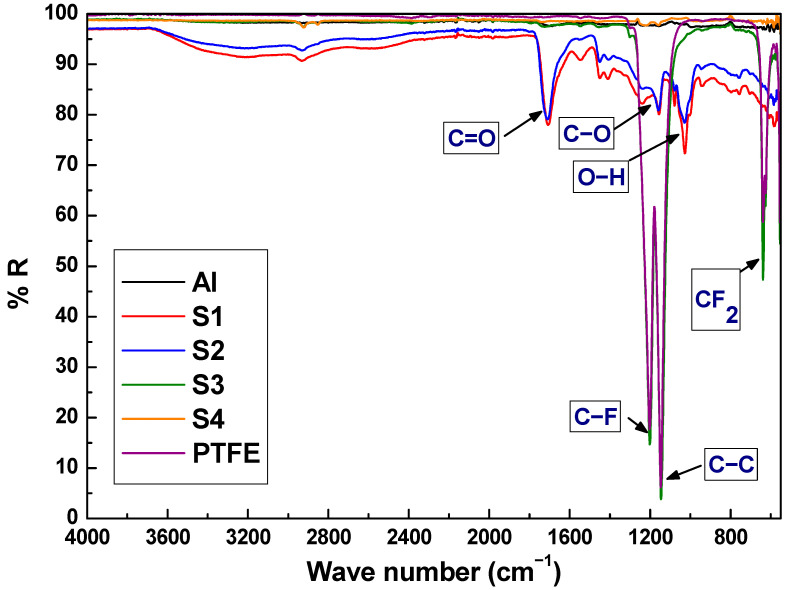
FTIR spectra of the S1, S2, S3, S4 and a reference PTFE sample.

**Figure 5 polymers-14-04356-f005:**
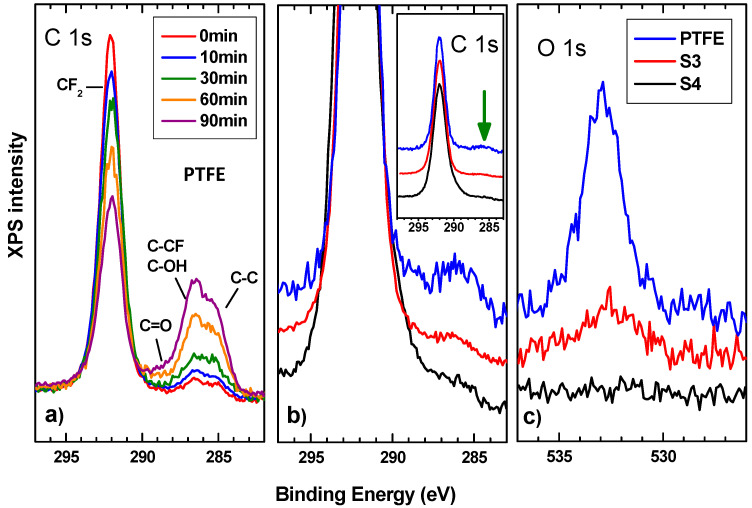
(**a**) XPS spectra of the C 1s corresponding to the PTFE reference sample for different irradiation times. (**b**) Right: XPS spectra of the C 1s corresponding to the PTFE, S3, and S4 samples together. Left: Zoom in to emphasize spectra differences. Spectra were normalized to the maximum intensity to highlight line shape differences, which provides direct valuable insight regarding the chemical environment of C atoms. (**c**) XPS spectra of the O 1s corresponding to the PTFE, S3 and S4 samples together.

**Figure 6 polymers-14-04356-f006:**
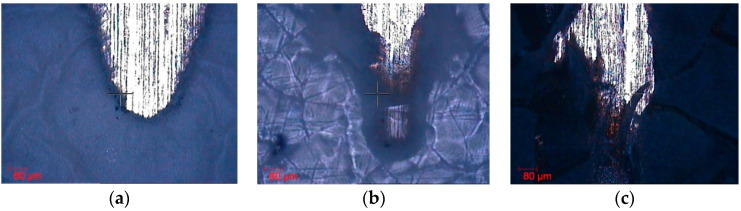
Optical images show the surface trace of the samples S2 (**a**), S3 (**b**), and S4 (**c**) at the scale of 80 µm, when the critical load is reached.

**Figure 7 polymers-14-04356-f007:**
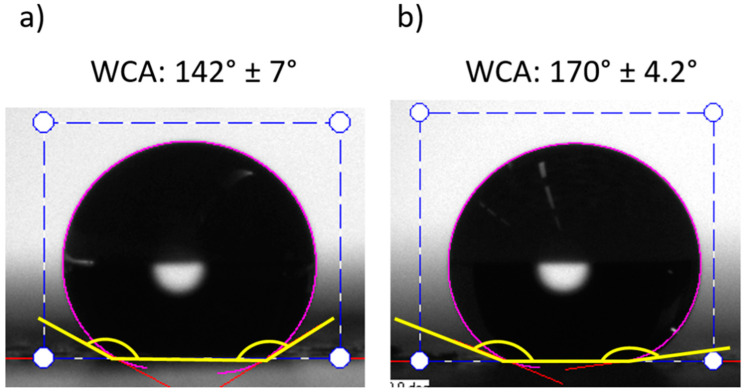
Static water contact angles (WCA) of samples S3 (**a**) and S4 (**b**). The sample S3 (**a**) exhibits a hydrophobic behavior (90° < WCA < 150°) and sample S4 (**b**) a superhydrophobic behavior (WCA > 150°).

**Figure 8 polymers-14-04356-f008:**
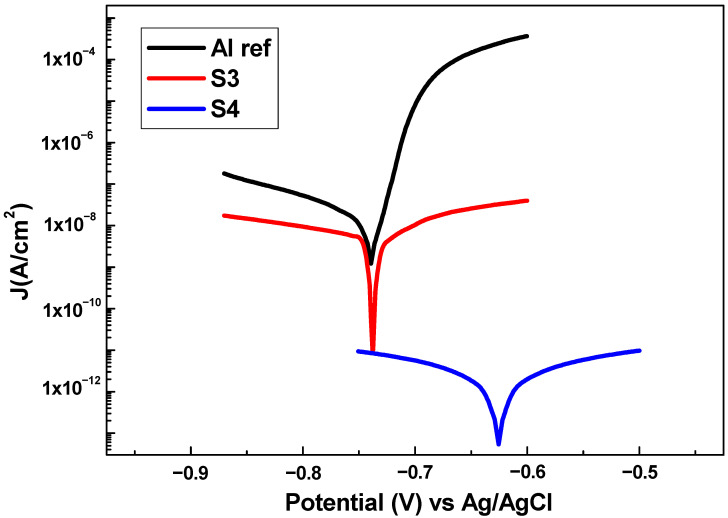
Tafel plots corresponding to the aluminum bare substrate and the different aluminum samples composed of PTFE composite being tested in 6 wt% NaCl aqueous solution.

**Table 1 polymers-14-04356-t001:** Summary of the methods, composition and heat treatments of Sx samples.

Sample (Sx)	Composition	Procedures	Heat Treatment (HT)		
S1	PAA + β-CD	ES		Temp (°C)	Time (min)
S2	PAA + β-CD	S1 + HT_1_	HT_1_	220	40
S3	PAA + β-CD + PTFE-1	S2 + SC_1_ + HT_2_	HT_2_	260	60
S4	PAA + β-CD + PTFE-1,2	S3 + SC_2_ + SP + HT_3_	HT_3_	340	80

**Table 2 polymers-14-04356-t002:** Configuration of the electrospinning and electrospraying parameters.

Parameters	ES	SP
Applied voltage (needle/collector) (KV)	3.55/−3.05	8.90/−3.00
Flow rate (mL/h)	1.5	0.5
Deposition time/sample (min)	30	25
Distance	15	15

**Table 3 polymers-14-04356-t003:** Summary of coating thickness, average fiber diameter and surface roughness.

Sample (Sx)	Thickness (µm)	Df (µm)	Ra (µm)
S1	37 ± 3	5.90 ± 0.45	1.05 ± 0.20
S2	64 ± 4	3.09 ± 0.24	6.87 ± 0.99
S3	81 ± 3	――	0.97 ± 0.25
S4	111 ± 5	――	5.58 ± 0.91

**Table 4 polymers-14-04356-t004:** Scatch critical loads of S2, S3, and S4 samples.

Sample (Sx)	Scratch Resistance (N)
S1	――
S2	1.0 ± 0.1
S3	3.3 ± 0.1
S4	8.2 ± 0.2

**Table 5 polymers-14-04356-t005:** The roll-off water angles ($\alpha_{\mathrm{roll}-\mathrm{off}}$), the advancing contact angle ($\theta_{\mathrm{adv}}$), the receding contact angle ($\theta_{\mathrm{rec}}$), and the contact angle of hysteresis (CAH) of the S3 and S4 samples.

Angles	S3	S4
$\alpha_{\mathrm{roll}-\mathrm{off}}$	>60°	<5°
$\theta_{\mathrm{adv}}$	123°	154°
$\theta_{\mathrm{rec}}$	78°	152°
CAH	45°	2°

**Table 6 polymers-14-04356-t006:** Summary table of the Tafel analysis parameters obtained for uncoated aluminum substrate (6061-T6) and composite coatings (Sx). All the samples of this study have been tested in 6 wt% NaCl aqueous solution.

Sample	Jcorr (nA/cm^2^)	Ecorr (V)	Corrosion Rate (nm/year)	βa (V/dec)	βc (V/dec)	η (%)
Al ref	217.320	−0.76	2427.20	0.02	0.13	0
S3	50.468	−0.73	563.66	0.11	0.16	76.77
S4	0.040	−0.62	0.45	0.30	0.29	99.98

## Data Availability

Not applicable.

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
