# Peer review of "Novel Design of Superhydrophobic and Anticorrosive PTFE and PAA + β − CD Composite Coating Deposited by Electrospinning, Spin Coating and Electrospraying Techniques"

_polymers, 2022, doi:10.3390/polym14204356_

Round 1
Reviewer 1 Report
Abstract: All acronyms used in the publication for the first time have to be explained (PTFE, PAA+β-CD, CAH)
Introduction: The comparison of other examples of electrospraying PTFE nanoparticles should be added. Moreover, the novelty of the work and the application of the obtained materials have to be clearly presented.
2.2.1. Solution and dispersion preparation: The composition of the dispersed PTFE-1 has to be added.
What does “profile rugosimeter” mean?
The images obtained using the confocal microscope and a field- emission scanning electron microscope have to be shown. Moreover, the three-dimensional (3D) images of the surfaces, obtained by means of Wyco RST-500 interferometric profilometer (Veeco Corporate, Plainview, NY, USA) using the vertical scanning interferometry (VSI), mode have to be included.
FTIR of PTFE has to be added. Moreover, an explanation is also necessary as to why in the case of S1 and S2 there is no difference in the FTIR spectra, and why the differences between S3 and S4 spectra are so significant. Describe the differences and explain.
Figure 6. The water contact angles have to be marked. In the present form, the differences are illegible.
Reviewer 2 Report
The manuscript is dedicated to the creation of superhydrophobic coatings based on PTFE. The article is interesting, the authors have done a great job, but the manuscript needs to be improved:
1) Introduction needs to be rewritten:
a) the main and most interesting achievement in the article is the obtaining of SH coatings with high strength (high adhesion for PTFE is very good) and corrosion resistance. However, the authors practically do not mention this in the introduction.
b) there is no comparison with the works already available in the scientific community (on the SH and PTFE tags in the scopus of 408 documents).
c) in the title, the authors indicate PAA + B-CD, but this is not in the introduction and purpose of the work.
2) The authors managed to achieve good results, but the process includes 4 stages (+heating):
a) why is PAA+B-CD used? (texturing only?)
b) Figure 3 shows that SC leads to a decrease in surface microrelief. Why do the authors necessarily use SC in the S4 sample (maybe simpler ES + HT + SP + HT)?
3) Paragraph 3.3. An increase in roughness almost always leads to a decrease in the strength of the coating. As confirmation, more detailed results with photomicrographs after loading should be given.
4) What is the coating thickness after each stage?
5) It is necessary to write in more detail the procedure for measuring SA. In the scientific literature there are works devoted to the study of the influence of conditions on the roll angle.
6) Table 6. What do columns 5 and 6 mean? There is no description in the text.
I hope my comments will be useful to the authors for finalizing the manuscript.
Round 2
Reviewer 2 Report
The authors managed to significantly improve the manuscript. In the introduction there is a comparison with previously described works and a description of the work presented.
The article can be recommended for publication. The only thing I would like to add is that it is not entirely correct to talk about an increase in adhesion when comparing a smooth and rough surface. When a developed hierarchical structure appears on the surface (which is a necessary condition for achieving superhydrophobicity), the key factor affecting the lyophilic properties of the surface will be the preservation of this structure. For example, good adhesion of the coating to the substrate will not clearly show the retention of superhydrophobicity under mechanical stress. Therefore, in addition to testing for adhesion, it is worthwhile to conduct a study on the resistance of superhydrophobic properties to abrasion.
Regardless, good job!